# Bloodstream Infections Caused by Carbapenem-Resistant Pathogens in Intensive Care Units: Risk Factors Analysis and Proposal of a Prognostic Score

**DOI:** 10.3390/pathogens11070718

**Published:** 2022-06-23

**Authors:** Giorgia Montrucchio, Andrea Costamagna, Tommaso Pierani, Alessandra Petitti, Gabriele Sales, Emanuele Pivetta, Silvia Corcione, Antonio Curtoni, Rossana Cavallo, Francesco Giuseppe De Rosa, Luca Brazzi

**Affiliations:** 1Department of Surgical Sciences, University of Turin, 10126 Turin, Italy; andrea.costamagna@unito.it (A.C.); tommaso.pierani@gmail.com (T.P.); petitti.alessandra@libero.it (A.P.); g.sales@unito.it (G.S.); luca.brazzi@unito.it (L.B.); 2Department of Anesthesia, Intensive Care and Emergency, Città Della Salute e Della Scienza di Torino University Hospital, 10126 Turin, Italy; 3Department of General and Specialized Medicine, Division of Emergency Medicine and High Dependency Unit, Città Della Salute e Della Scienza di Torino University Hospital, 10126 Turin, Italy; emanuele.pivetta@unito.it; 4Department of Medical Sciences, Infectious Diseases, University of Turin, 10126 Turin, Italy; silvia.corcione@unito.it (S.C.); francescogiuseppe.derosa@unito.it (F.G.D.R.); 5Division of Geographic Medicine, Tufts University School of Medicine, 145 Harrison Ave, Boston, MA 02111, USA; 6Microbiology and Virology Unit, Città Della Salute e Della Scienza di Torino University Hospital, 10126 Turin, Italy; acurtoni@cittadellasalute.to.it (A.C.); rossana.cavallo@unito.it (R.C.)

**Keywords:** carbapenem, multidrug resistance, antimicrobial resistance, Gram-negative bacteria, bloodstream infection, sepsis, risk factors, critically ill, intensive care

## Abstract

Considering the growing prevalence of carbapenem-resistant Gram-negative bacteria (CR-GNB) bloodstream infection (BSI) in intensive care units (ICUs), the identification of specific risk factors and the development of a predictive model allowing for the early identification of patients at risk for CR-*Klebsiella pneumoniae*, *Acinetobacter baumannii* or *Pseudomonas aeruginosa* are essential. In this retrospective case–control study including all consecutive patients showing an episode of BSI in the ICUs of a university hospital in Italy in the period January–December 2016, patients with blood culture positive for CR-GNB pathogens and for any other bacteria were compared. A total of 106 patients and 158 episodes of BSI were identified. CR-GNBs induced BSI in 49 patients (46%) and 58 episodes (37%). Prognosis score and disease severity at admission, parenteral nutrition, cardiovascular surgery prior to admission to ICU, the presence of sepsis and septic shock, ventilation-associated pneumonia and colonization of the urinary or intestinal tract were statistically significant in the univariate analysis. The duration of ventilation and mortality at 28 days were significantly higher among CR-GNB cases. The prognostic model based on age, presence of sepsis, previous cardiovascular surgery, SAPS II, rectal colonization and invasive respiratory infection from the same pathogen showed a C-index of 89.6%. The identified risk factors are in line with the international literature. The proposal prognostic model seems easy to use and shows excellent performance but requires further studies to be validated.

## 1. Introduction

Carbapenem-resistant Gram-negative bacteria (CR-GNB), including *Enterobacterales*, *Acinetobacter baumannii* and *Pseudomonas aeruginosa*, have spread alarmingly in recent years [1], leading to their inclusion in the World Health Organization (WHO) global critical priority list of antibiotic-resistant bacteria [2]. These CR-GNB carry a risk of morbidity and mortality higher than their carbapenem-susceptible counterparts [3,4,5,6]. Bloodstream infections (BSI) due to carbapenem-resistant bacteria are usually diagnosed in subjects with severe underlying conditions, in the critically ill and/or in immunocompromised patients [7]. Isolates are generally resistant to first-line antibiotics resulting in a high rate of initial inappropriate therapy and/or leading to the use of less effective and more toxic drugs. This explains the frequent use of antibiotic combinations regimens with controversial results in terms of efficacy, toxicity and collateral environmental damage. Knowledge of local epidemiology, etiological distribution of causative agents and antibiotic resistance and factors associated with poor patient outcome are pivotal information to guide infection control and antimicrobial stewardship policies and to assist clinicians to choose the best therapeutic approach [8]. Other risk factors included the use of invasive devices and procedures, such as mechanical ventilation and catheterization, and exposure to the hospital environment, particularly the intensive care unit (ICU). Interventions to mitigate CR-GNB infections should target these factors [9].

The treatment of BSI caused by these strains is challenging due to their high morbidity and mortality rates [10,11]. In severe infections, with a high risk to evolve into septic shock, a delayed adequate antimicrobial therapy is associated with an increased mortality. The rapid identification of the aetiological agents of infection is essential for the administration of an early and appropriate antimicrobial therapy [12]. Unfortunately, the median time-to-positivity of blood cultures (BC) is frequently long, especially in patients with GNB BSI where it ranges from 5.9 to 49.8 h [13]. Although in recent years new microbiological assays have had to speed up the diagnostic process [14], additional time is necessary to obtain species identification and antimicrobial susceptibility test (AST) results [15]. For these reasons, clinical evaluation remains the milestone to ensure an early and appropriate initiation of empirical broad-spectrum antimicrobial therapy through the identification of those patients who have the highest risk of multidrug resistance, especially CR, BSI.

A growing number of experiences approached the development of scores based on patient risk factors, with the aim of the early recognition of patients eligible for an effective empirical therapy against the CR pathogens. In the different studies, specific pathogens, antimicrobial-resistance patterns and/or specific subpopulations of patients were considered [16,17,18,19,20]. Many studies have been conducted in Italy [21,22,23], a country characterized by one of the highest rates of resistance to antibiotics in Europe [24,25]. Unfortunately, despite the importance of this aspect, experiences with critically ill patients in ICU are limited [26,27]. With this premise, we decided to analyse a retrospective cohort of patients affected by BSI and recruited in five ICUs of a University Hospital in Italy with the aim of describing the current distribution of the epidemiological, etiological and susceptibility pattern of CR-GNB and to develop a risk-assessing tool to predict carbapenem-resistant aetiology in critically ill patients with BSI admitted in ICU.

## 2. Methods

### 2.1. Study Design

The present multicentric cohort study was conducted in five ICUs (general respiratory, emergency, cardiac surgical, post-surgical and neurosurgical) of the ‘Città della Salute e della Scienza’ University Hospital in Turin (Italy) between January 2016 and December 2016. Data were collected retrospectively and anonymously from medical records and from the laboratory information system by three investigators (T.P., A.P. and V.R.) and revised and checked by two senior investigators (G.M. and A.C.). The local Ethics Committee approved the study protocol (n. 0042680). Due to the retrospective nature of the study, written consent to participate was waived.

### 2.2. Subjects

All patients with at least one positive blood culture (BC) taken at least after 48 h after ICU admission were enrolled [28]. False positive blood culture, defined according to the ECDC criteria [29,30], were excluded.

Patients were divided into two groups: (1) CR-GNB group-patients developing at least one positive BC to CR-GNB belonging to the following species: *Klebsiella pneumoniae* and/or *Acinetobacter baumannii* and/or *Pseudomonas aeruginosa*. (2) Non-CR-GNB group-patients developing at least one positive BC to any pathogen other than those described for the CR-GNB group.

### 2.3. Variables Collected and Evaluated to Identify Potential Risk Factors

(1)Age and gender;(2)Ward of initial admission, the occurrence of cardiovascular surgery before ICU admission, type of ICU;(3)Prior antimicrobial therapy (ATB) therapy;(4)ICU prognostic and clinical severity scoring systems at admission (APACHE score [31], SAPS II score [32], SOFA score [33], CHARLSON index [34]);(5)Life support measures (mechanical ventilation, tracheostomy, extracorporeal life support, renal replacement therapy, parenteral nutrition, left ventricular assist device and pacemaker/internal cardioverter and defibrillator);(6)BSI (central venous catheter site for BC positivity, number of BSI episodes);(7)Laboratory tests at positive BC timepoint;(8)Comorbidities: main diagnosis for ICU admission (i.e., any type of shock, cardiovascular, respiratory, abdominal or miscellaneous-other), sepsis or septic shock occurrence;(9)Concurrent infection and/or colonization as potential BSI source: ventilator-associated pneumonia (VAP), ventilator-associated tracheobronchitis (VAT), urinary tract (UT) colonization, rectal colonization, central line-associated bloodstream infection (CLABSI).

### 2.4. Variables Collected to Describe Therapeutic Measures and Outcome

(1)Measures implemented after BSI evidence (i.e., CVC removal, empiric ATB match with antibiogram, need for proper ATB therapy switch after antibiogram evidence);(2)Measures of outcome (i.e., ventilator-free days, ICU and in-hospital length of stay, in-hospital death and 28 days death).

### 2.5. Definitions

Resistance to carbapenem was defined, according to the Centers for Disease Control and Prevention (CDC), as follows:− *K. pneumoniae*-defined as ‘carbapenem resistant’ when found resistant (R) to at least 1 of the following: imipenem, meropenem, doripenem, ertapenem;− *P. aeruginosa* and *A. baumannii*-defined as ‘not susceptible to carbapenems’ when found to have an intermediate sensitivity (I) or to be resistant (R) to at least 1 of the following: imipenem, meropenem, doripenem.

A new manifestation of BSI due to the same pathogen was defined as a new positive BC after a 14-day course of effective ATB and in presence of clinical data and laboratory tests (leukocytes, C reactive protein and procalcitonin) compatible with the resolution of the infection and an effective source control [29]. Empirical therapy was defined as the administration of antibiotics before the availability of the sensitivity report. Appropriate empirical therapy was defined as a therapy containing at least one in vitro active drug (according to the susceptibility pattern of the isolate) administered within 24 h form the index BCs. Inappropriate empirical therapy was defined as inactive antibiotic administration or delayed antibiotic therapy. Definitive antibiotic therapy was defined as the antibiotic treatment administered according to susceptibility results. Duration of antibiotic treatment was defined as the number of consecutive days during which the patient received an appropriate antibiotic regimen. Source control was defined as performing any non-surgical or surgical procedure to treat the infection source at any site including, among others, the urinary tract, biliary tract and surgical site, and the removal of any device deemed as the source of BSI within 7 days of index BCs.

### 2.6. Microbiology

Surveillance cultures are routinely performed at ICU admission and once a week and include tracheal aspirate, urinary cultures and rectal swab screening for CR-GNB.

Suspected GNB colonies grown on MacConkey II Agar (Becton–Dickinson, Franklin Lakes, NJ, USA) from tracheal and urinary cultures were identified and tested for CR using the semi-automated Microscan Walkaway 96 plus System (Beckman Coulter, Indianapolis, IN, USA) with Neg Combo 83 (Beckman Coulter) panels and a complete antimicrobial susceptibility test (AST) was provided. GNB isolated from rectal swab on selective solid media, Brilliance CRE AGAR (Oxoid, Basingstoke, UK), were identified by MALDI-TOF VITEK MS (bioMérieux, Marcy-l’Étoile, France) and CR was confirmed by meropenem and imipenem E-test (bioMérieux).

BCs are performed in case of clinical suspicion of infection. BCs were incubated using the automated BC system BACT/ALERT 3D (bioMérieux). Positive BC were processed according to the laboratory routine workflow as previously described [35]. Briefly, BC were seeded on the appropriate solid medium after Gram staining results (Columbia Agar with 5% Sheep Blood, MacConkey II Agar, Columbia CNA Agar with 5% Sheep Blood, Chocolate Agar; Becton–Dickinson, USA) and isolated colonies after overnight incubation were tested for identification and AST both on Microscan Walkaway 96 plus System (Beckman Coulter). The tested carbapenems minimum inhibitory concentrations (MIC) were 0.5–1 μg/mL for ertapenem, 1–8 μg/mL for imipenem, 0.12, 2, 8 μg/mL for meropenem with Neg Combo 83 (Beckman Coulter) panels and 0.002–32 μg/mL for both imipenem and meropenem with E-test (bioMérieux). For all tested antibiotics MIC values were interpreted according to the EUCAST clinical breakpoints. [36]

### 2.7. Endpoints

The primary endpoint was to develop a prognostic model capable of identifying, among the patients with positive blood cultures, those with BSI caused by CR-GNB species. Patients with BSI caused by CR-*Acinetobacter baumannii*, *Klebsiella pneumoniae*, *Pseudomonas aeruginosa* were compared to those in whom BSI was due to susceptible bacteria. A logistic regression to identify predictive factors of CR bacteria was performed. A prognostic score for the risk of CR-GNB BSI was developed. The secondary endpoints were to evaluate the impact of CR-GNB BSI on patients’ outcomes.

### 2.8. Statistical Analysis

Descriptive data were tested for normal distribution by the Shapiro–Wilk test and presented as mean (±standard deviation [SD]) or median (with interquartile range [IQR]) as appropriate. Wilcoxon rank-sum test for unmatched samples was used for non-parametric continuous variables. Categorical variables were analysed with Chi-squared or Fisher’s exact test, as appropriate. All study variables showing a significant difference (*p* < 0.05) between CR-GNB and non-CR-GNB groups, were included in a univariate logistic regression to assess their role as risk factors for CR-GNB BSI. Multivariate logistic regression was performed using a selection of variables showing statistical significance at univariate analysis (*p* < 0.05). In the case of indicators described by different variables (severity upon admission to the ICU, presence of sepsis or septic shock, possible starting sources of BSI), the variable included in the model was that associated with a higher statistical value at the univariate. For the predictive score, the variables were selected according to international literature and univariate logistic models and combined in a prognostic model whose discrimination and calibration were assessed. Discrimination ability was measured using the C-index. Calibration was visually assessed by evaluating the predicted and observed the probability plot of infection after bootstrapping at 1000 replicates. Statistical analyses were conducted using Stata 16.1/SE (Stata Corp TX, College Station, TX, USA) and R version 3.6.3 (The R Foundation for Statistical Computing, 2020).

## 3. Results

During the study period, data from 106 patients with BSI were collected, for a total of 158 BSI episodes. At least one blood culture positive for CR-GNB pathogens was identified in 49 (46%) patients. Overall, 76 (72%) patients had a single episode and 30 (29%) patients experienced multiple BSI episodes. Among the 30 patients experiencing multiple BSI episodes, 3 (6% of the total) presented BSI only from CR-GNB pathogens, 11 (19%) only from non-CR GNB pathogens and 16 patients (33%) presented different episodes caused by both non-CR GNB and CR GNB pathogens. The percentage of patients with multiple episodes was significantly higher among patients with at least one episode of CR-GNB BSI (*p* = 0.026) (Figure 1; Table 1 and Table 2). In CR-GNB group, registered MIC were always >32 μg/mL when confirmed by E-test (bioMérieux) methods.

BSIs were caused in the CR group mostly by CR-*K. pneumoniae* (49; 88%), followed by CR-*A. baumannii* (4; 8%) and by CR-*P. aeruginosa* (2; 4%).

Microbiological isolates (*n* = 182) are shown in Figure 1.

### 3.1. Baseline Characteristics and Risk Factors

There were not differences in age (*p* = 0.7998) and gender (*p* = 0.4260) in the study groups. APACHE (*p* = 0.0121), SAPS II (*p* = 0.0101) and SOFA (*p* = 0.0305) scores at ICU admission were significantly higher in the CR-GNB group (Table 2).

The percentage of tracheostomized patients was higher in the CR-GNB group (*p* = 0.0080) but the use of mechanical ventilation did not differ among groups (*p* = 0.4930). Parenteral nutrition administration was significantly more frequent in the CR-GNB group (*p* = 0.0280). No differences were observed neither in terms of the type (*p* = 0.3790 for Carbapenem and *p* = 0.9120 for Fluoroquinolone) nor in terms of the duration (*p* = 0.6020) of antibiotic treatment before ICU admission.

Most patients were admitted to ICU from a surgical ward (65; 59%). The CR-GNB group was characterized by a higher frequency of previous cardiac and/or vascular surgery (30; 61% vs. 22; 39%).

Sepsis (*p* = 0.0010) and septic shock (*p* = 0.0450) at ICU admission were most frequent in the CR-GNB group. While the rates of VAP (*p* = 0.0020), VAT (0.0030), urinary tract (*p* = 0.0340) and rectal colonization (*p* < 0.001) were significantly higher in the CR-GNB group, the incidence of CLABSI did not differ between groups (*p* = 0.3570).

### 3.2. Therapeutic Approach and Outcomes

There was no difference between the two groups either in the management of CVC (*p* = 0.1510) or ATB revision (*p* = 0.9150) after the BSI finding. A correspondence between the empiric ATB and the antibiogram indications emerged more frequently in the non- CR-GNB group (*p* = 0.0080) (Table 3).

Ventilator-free days were higher (*p* < 0.001) in the non-CR-GNB group (19 (IQR: 0–25)) than in the CR-GNB group (0 (IQR: 0–4)). Even in the presence of a statistically non-different in-hospital mortality (*p* = 0.0680) between the two groups (24 (49%) vs. 18 (32%)), the rate of 28-day mortality was statistically lower (*p* = 0.0430) in the non-CR-GNB group (10; 18% vs. 17; 35%).

Infections were considered the leading cause of death either in the overall population (32; 80%) or in the two study groups (15; 88% non- CR-GNB vs. 17; 74% CR-GNB group, *p* = 0.2630).

### 3.3. Logistic Regression

Logistic regression model, in which five variables (SAPS II score, parenteral nutrition, cardio-vascular surgery before ICU, sepsis and rectal colonization) were included, highlighted that SAPS II score (OR 1.04, 95% CI 1.00–1.09), cardio-vascular surgery before ICU admission (OR 3.34, 95% CI 1.20–9.29) and rectal colonization (OR 10.58, 95% CI 3.59–31.55) are all factors statistically associated with BSI from CR-GNB (Table 4).

### 3.4. Prognostic Model 

The prognostic model was realized using the following variables defined on the basis of literature, clinical experience and data obtained from the univariate model: age, presence of sepsis, previous cardiovascular surgery in the same hospital, the SAPS II score, rectal colonization by the same pathogen, invasive respiratory infection by the same pathogen. The prognostic nomogram and the calibration of the model (C-index 0.896) are shown in Figure 2 and Figure 3.

Figure 2 is the nomogram built using the variables included in the model. Their presence or absence allow an overall probability of developing CR-GNB BSI to be obtained. Such a model has been calibrated based on a 1000-repetition bootstrap process (see Methods), and Figure 3 shows the results of it. The prediction model has some divergences from the ideal prediction (i.e., the dotted line) but their entity is negligible at about 40% and 90% probability of CR-GNB BSI.

## 4. Discussion

Our data show that in a high rate of cases (46%) the aetiologic agents of BSI are CR-GNB as confirmed by the international literature, where the growing impact of Gram-negative aetiology in BSI is reported to be around 50% [37]. More in detail, it appears that around 80% of the cases are attributable to *K. pneumoniae*, 8% to *A. baumannii* and 4% to *P. aeruginosa*. In Italy, *E. coli*, *K. pneumoniae*, *P. aeruginosa* and *A. baumannii* together constitute 73% of BSI agents and carbapenem resistance rate is 13.1% in GNB-BSI and 16.7% in hospital acquired infections [38,39].

In our cohort, among invasive infections, *K. pneumoniae* was the most frequent CR-GNB (22.6% of isolated, carbapenem resistance rate of 37%), followed by *A. baumannii* (7.1% of all isolated, but with resistance rate to carbapenems of 79.9%) and *P. aeruginosa* (12.4% of all isolated, carbapenem resistance rate of 28%). It is worth noting that the population enrolled in our cohort is relatively young and is characterized by a high clinical severity, as demonstrated by the prognostic score values. The APACHE II, SAPS II and SOFA scores calculated at ICU admission were significantly higher among CR-GNB cases, confirming that this aetiology mainly affects critically ill patients [9,40].

An interesting observation is that the CR-GNB group is characterized by a higher percentage of patients undergoing cardiac or vascular surgery in the 30 days prior to ICU admission (61% vs. 39% of the control group), suggesting a possible role of surgery in predicting the occurrence of CR-GNB aetiology, in line with previous studies [22,41,42].

Other sources of potentially BSI-associated infections included ventilator-associated pneumonia (VAP), ventilator-associated tracheobronchitis (VAT), urinary tract and gastrointestinal colonization which were all significantly more frequent among CR-GNB group (VAP 38.8% vs. 12.3%-VAT 35% vs. 11%). This data is also in line with the literature, where respiratory infections are reported as the most common source of pathogen isolation non-susceptible to carbapenems [37,43].

Our data support the hypothesis that CR-GNB carriage status is strongly associated with the risk of BSI. In the multivariate analysis, gastrointestinal colonization was confirmed to be a strong independent risk factor for CR-GNB BSI (OR of 10.58, 95% CI 3.59–31.55). This is consistent with literature evidence that intestinal colonization by CRE is a significant risk factor for the development of invasive infection in critically ill patients admitted to ICU [20,40,44,45]. *P. aeruginosa* also frequently colonizes the gut of hospitalized patients and its colonization is a significant risk factor for the development of infection, as well as for *A. baumannii* [46,47,48].

With regard to previous antibiotic therapies, there was no correlation between the duration of antimicrobial therapy, nor between the administration of antibiotics known to select more resistances, such as carbapenems or fluoroquinolones, and the carbapenem-resistant BSI.

Inappropriate empiric antibiotic therapy was administered more frequently in the group of patients with CR-GNB BSI, since 77% of these patients received a therapy that did not contain any of the active antibiotics according to the antibiogram, compared to the 53% in non-CR-GNB controls. This is also in line with other studies where inappropriate empiric antibiotic therapy is prescribed up to 70% of cases of multidrug-resistant bacteria, and it is recognized as one of the risk factors for mortality [39,49,50,51].

CR-GNB patients were treated with mechanical ventilation for a longer period. This can be interpreted as a sign of severity on the admission and of the greater impact of the infection. Always in line with the literature, sepsis and septic shock were in fact present, respectively, in 88% and 47% of CR-GNB patients compared to 60% and 28% of controls [52].

Overall, 28-day mortality was 25%, but it reached approximately 35% among patients who developed CR-GNB infection. The data confirmed the known higher mortality associated with CR-aetiology than the one compared to that due to susceptible Gram-negative infections, whose frequency varies from 30% to more than 70% [20,22,40,49,52].

Specific predictors of CR-aetiology in critically ill ICU patients with BSI, such as SAPS II score, cardio-vascular surgery before ICU admission, gastro-intestinal colonization, were identified. These factors were combined with literature findings and weighted to develop a simple tool to be used by clinicians to estimate the probability of CR-GNB aetiology in patients with suspected BSI. Each considered variable (age, presence of sepsis, previous cardio-vascular surgery in the same admission, SAPS II, gastro-intestinal colonization or invasive respiratory infection by the same pathogen) corresponds to a specific relative risk score that can be combined with the others, based on the patient’s characteristics, in a well-calibrated model (C-index = 89.6%).

Several scores have been proposed in order to estimate the risk of developing infections by multi-resistant pathogens [18,19,22,26,27,37]. However, these scores were focused either on different settings from ICU, or not only on BSI or on specific subgroups of patients. For instance, Giannella et al. [19,20] investigated the risk in liver transplant recipients colonized with carbapenem-resistant Enterobacteriaceae; Tumbarello et al. [22] identified risk factors specifically associated with *K.pneumoniae* KPC infection–regardless on the site- in all hospitalized patients, while Vadesuvan et al. [26] proposed a simple but effective score to predict nosocomial resistant Gram-Negative Bacilli infections among ICU patients, already validated in an external cohort [27].

It is clear that the purpose of a score is the early identification of patients requiring an empirical therapy capable of covering CR-pathogens. As previously underlined, ensuring an adequate early antibiotic therapy is necessary especially in the context of ICU patients; it is equally important to limit the use of broad-spectrum antibiotics as much as possible, as recommended by the antimicrobial stewardship programs [53,54]. In addition, the times for returning the microbiological results differ strongly from centre to centre, especially on the basis of the routinely used diagnostic techniques. Despite the recent introduction of rapid techniques to recognize the BSIs, such as MALDI-TOF or other molecular methods, which allow for the rapid identification of the microbial species and their principal resistance genes on the day of BC positivity [14], the time-to-results remains a critical issue mostly for complete antimicrobial susceptibility test reports.

Considering all those issues, it is essential to define-on the basis of the risk factors, the recent history and the current clinical presentation-which patients should be referred to a so-called “pre-emptive” therapy against CR-pathogens [39,42]. The proposed score seems to have a good calibration on the studied cohort. A low risk according to our score can reassure the clinician of the lack of need for such coverage; conversely, a higher score—as defined by the likely CR-GNBs isolation in blood culture—can lead to considering a CR-covering antimicrobial therapy earlier (ideally, at the time of blood culture). Since the delayed administration of an effective antibiotic therapy is the strongest predictor of poor survival outcomes [55], a tool facilitating an early adequate therapy becomes essential for the clinician.

### Limits

Our study has limitations. First, it is a retrospective study and recent, rapidly evolving scenarios might have changed the situation, especially considering the pandemic and the introduction of new antibiotics for multi-resistant pathogens. Moreover, the retrospective nature of the study did not allow us to properly assess the percentage of BSIs related to the presence of a central venous catheter, which is important data in the ICU setting. Another limiting factor is that the microbiology ecology may be shaped by local issues and the results could be influenced by the epidemiology of a restricted area of our country, even if we analysed a cohort of patients from five different ICUs. Additionally, our cohort of patients is from large tertiary teaching hospitals reflecting the complexity and epidemiology of patients managed in similar institutions. Italy is a country with a high prevalence of multidrug-resistant organisms, and our findings cannot be generalized to other countries with a lower prevalence of antibiotic resistance. Regarding the impact of previous antimicrobial use on the possibility to develop CR-GNB strains, although neither carbapenems nor fluoroquinolones seem to play a causative role, it was not possible to fully collect data on other antibiotic therapies. Finally, the number of BSI episodes due to CR-GNB strains was relatively small.

## 5. Conclusions

About a half of the patients enrolled in our study developed CR-GNB BSI, in clear superiority if compared to the literature and with a strong impact on patient survival, as these pathogens were associated with higher mortality and lower efficacy of initial empirical therapy. Specific risk factors—clinical severity of the patients, presence of sepsis, concomitance of respiratory infections and gastrointestinal colonization—are in line with the literature.

Our clinical easy, fast-performing prediction tool could enable a better targeting of the early antimicrobial treatment for BSI, maintaining an antimicrobial stewardship perspective in the ICU context with its elevated incidence of CR resistance.

Further multicentric and prospective studies are needed to confirm our data and to validate the model and better define the impact of each risk factor.

## Figures and Tables

**Figure 1 pathogens-11-00718-f001:**
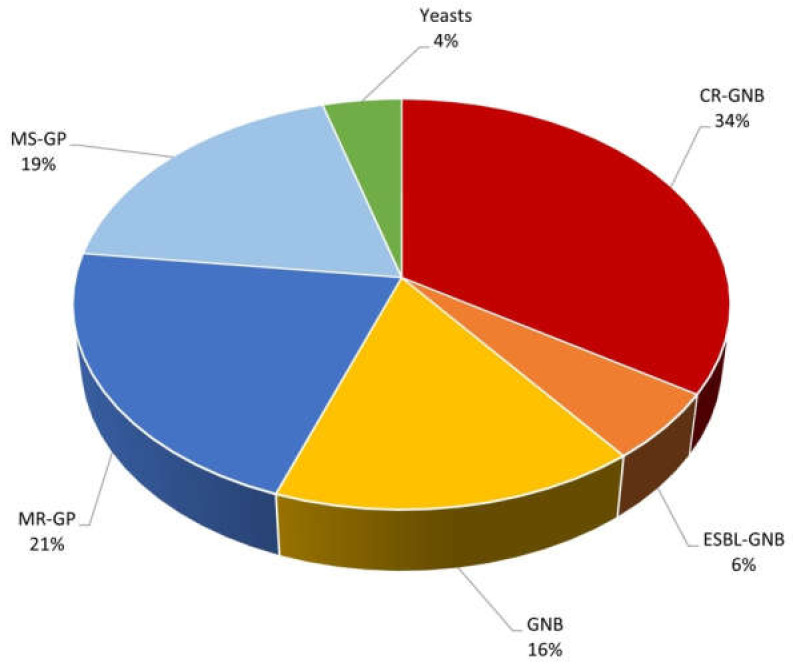
Microbiological isolates in the overall population. CR-GNB, Carbapenem-Resistant Gram-Negative Bacteria (*n* = 62); ESBL-GNB, Extended Spectrum Beta-Lactamase Gram-Negative Bacteria (*n* = 10); GNB, non-multi drug-resistant Gram-Negative Bacteria (*n* = 29); MR-GP, Methicillin-Resistant Gram-Positive Bacteria (*n* = 39); MS-GP, Methicillin-Sensitive Gram-Positive Bacteria (*n* = 34); Yeasts (*n* = 8).

**Figure 2 pathogens-11-00718-f002:**
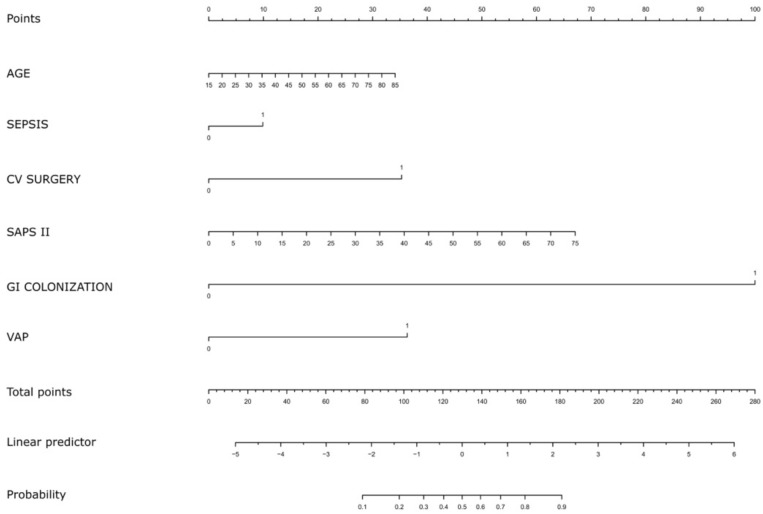
Nomogram for predicting risk of developing CR-GNB BSI. Points are summed for each risk factor. List of abbreviations: CR-GNB, Carbapenem-Resistant Gram-Negative Bacteria; CV, Cardiac and/or Vascular; GI, gastro-intestinal; VAP, Ventilator-Associated Pneumonia.

**Figure 3 pathogens-11-00718-f003:**
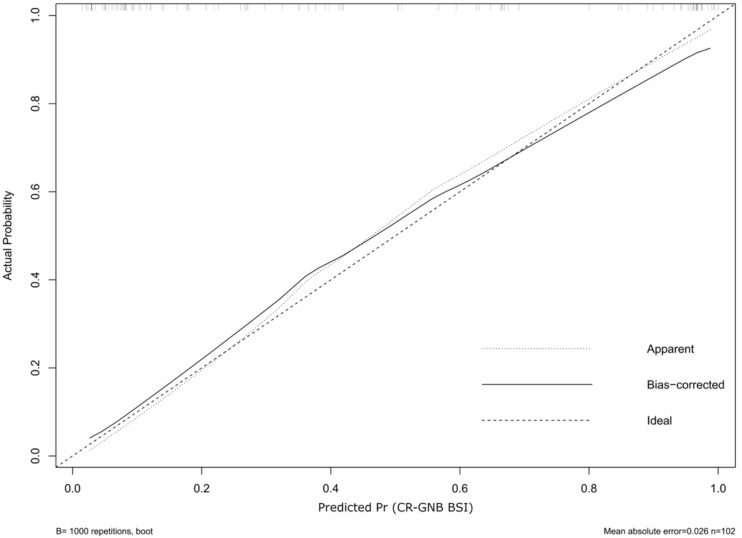
Calibration curves for the prognostic model. List of abbreviations: CR-GNB, Carbapenem-Resistant Gram-negative Bacteria.

**Table 1 pathogens-11-00718-t001:** Microbiological isolates species in the overall population (182 isolates, from 158 episodes in 106 patients).

Category	Species	Number (Overall%)
CR-GNB	*K. pneumoniae*	50 (27.5)
	*A. baumannii*	8 (4.4)
	*P. aeruginosa*	4 (2.2)
ESBL-GNB	*E. coli*	7 (3.9)
	*K. pneumoniae*	1 (0.6)
	*P. mirabilis*	2 (1.1)
GNB	*E. coli*	1 (0.6)
	*K. pneumoniae*	1 (0.6)
	*K. oxytoca*	3 (1.7)
	*E. aerogenes*	1 (0.6)
	*S. marcescens*	7 (3.9)
	*M. morganii*	3 (1.7)
	*P. aeruginosa*	5 (2.8)
	*S. maltophila*	1 (0.6)
	*C. koseri*	1 (0.6)
	*H. alvei*	1 (0.6)
	*A. xylosoxidans*	1 (0.6)
	*E. cloacae*	3 (1.7)
	*A. baumannii*	1 (0.6)
MR-GP	*S. epidermidis*	35 (19.2)
	*S. aureus*	2 (1.1)
	*S. haemolitycus*	2 (1.1)
MS-GP	*E. faecalis*	6 (3.3)
	*E. faecium*	8 (4.4)
	*S. epidermidis*	9 (5.0)
	*S. aureus*	4 (2.2)
	*S. haemolitycus*	1 (0.6)
	*S. capitis*	2 (1.1)
	*S. simulans*	1 (0.6)
	*G. sanguinis*	1 (0.6)
	*S. cohnii*	1 (0.6)
	*B. casei*	1 (0.6)
Yeasts	*C. albicans*	3 (1.7)
	*C. glabrata*	4 (2.2)
	*C. parapsilosis*	1 (0.6)
Total		182 (100)

**List of abbreviations:** CR-GNB, Carbapenem-Resistant Gram-Negative Bacteria; ESBL-GNB, Extended Spectrum Beta-Lactamase Gram-Negative Bacteria; GNB, non-multidrug-resistant Gram-Negative Bacteria; MR-GP, Methicillin-Resistant Gram-Positive Bacteria; MS-GP, Methicillin-Sensitive Gram-Positive Bacteria; Yeasts.

**Table 2 pathogens-11-00718-t002:** Baseline characteristics of patients and potential risk factors for CR-GNB BSI. List of abbreviations: CR-GNB, carbapenem-resistant Gram-negative bacteria; M, male; MV, mechanical ventilation; ECLS, Extra Corporeal Life Support; RRT, Renal Replacement Therapy; LVAD, Left Ventricular Assist Device; PM/ICD, Pacemaker/implantable cardioverter-defibrillator; BSI, Blood Stream Infection; CVC, Central Venous Catheter; CV, Cardiac and/or Vascular; PICC, Peripherally Inserted Central Catheter; VAP, Ventilator-Associated Pneumonia; VAT, Ventilator-Associated Tracheobronchitis; UT, Urinary Tract; CLABSI, Central Line-associated Bloodstream Infection.

Variable	Non-CR-GNB (57)	CR-GNB (49)	Total (106)	*p*-Value
*Demographics*				
Age (years)	64 (55–73)	65 (53–73)	64 (54–73)	0.7998
Gender M (N (%))	38 (67%)	29 (59%)	67 (63%)	0.4260
*ICU scoring systems*				
APACHE score	15 (12–20)	19 (15–23)	18 (14–21)	0.0121
SAPS II score	39 (28–49)	46 (38–54)	42 (33–52)	0.0101
SOFA admission	8 (5–9)	9 (7–11)	8 (5–10)	0.0305
CHARLSON index	4 (2–6)	4 (3–6)	4 (3–6)	0.4239
*Comorbidities*				
Main diagnosis				
Shock (N (%))	19 (33%)	18 (37%)	37 (35%)	0.865
Cardiovascular (N (%))	7 (12%)	8 (16%)	15 (14%)	
Respiratory (N (%))	18 (32%)	11 (23%)	29 (28%)	
Abdominal (N (%))	7 (12%)	7 (14%)	14 (13%)	
Other (N (%))	6 (11%)	5 (10%)	11 (10%)	
Sepsis (N (%))	34 (60%)	43 (88%)	77 (73%)	0.0010
Septic shock (N (%))	16 (28%)	23 (47%)	39 (37%)	0.0450
*Episodes of BSI*				
Single	46 (81%)	30 (61%)	76 (72%)	0.0260
Multiple				
non-CR-GNB first	11 (19%)	16 (33%)	27 (26%)	
CR-GNB first		3 (6%)	3 (3%)	
*Potential source of BSI*				
VAT (N (%))	6 (11%)	17 (35%)	23 (22%)	0.0030
UT colonization (N (%))	3 (5%)	10 (20%)	13 (12%)	0.0340
Rectal colonization (N (%))	3 (5%)	32 (65%)	35 (33%)	0.0000
CLABSI (N (%))	46 (81%)	36 (74%)	82 (77%)	0.3570
*Vital function support*				
MV (N (%))	46 (81%)	42 (86%)	88 (83%)	0.4930
Tracheostomy (N (%))	10 (18%)	20 (41%)	30 (30%)	0.0080
ECLS (N (%))	5 (9%)	10 (20%)	15 (14%)	0.1010
ECLS (days)	0 (0–0)	0 (0–0)	0 (0–0)	0.1471
RRT (N (%))	11 (19%)	14 (29%)	25 (24%)	0.2620
RRT (days)	0 (0–0)	0 (0–2)	0 (0–0)	0.2111
Parenteral nutrition (N (%))	14 (25%)	22 (45%)	36 (34%)	0.0280
LVAD (N (%))	5 (9%)	5 (10%)	10 (9%)	1.0000
PM/ICD (N (%))	9 (16%)	7 (14%)	16 (15%)	0.8290
*Prior ATB therapy*				
Duration of ATB therapy				
None	44 (77%)	35 (71%)	79 (75%)	0.602
<7 days	6 (11%)	7 (14%)	13 (12%)	
7–14 days	6 (11%)	4 (8%)	10 (9%)	
>14 days	1 (2%)	3 (6%)	4 (4%)	
Carbapenem before ICU admission (N (%))	6 (11%)	8 (16%)	14 (13%)	0.3790
Fluoroquinolone before ICU admission (N (%))	10 (18%)	9 (18%)	19 (18%)	0.9120

**Table 3 pathogens-11-00718-t003:** Therapeutic intervention following BSI diagnosis and measures of outcome. List of abbreviations: CR-GNB, carbapenem-resistant Gram-negative bacteria; CVC, Central Venous Catheter; ATB, Antibiotic; LOS, Length of Stay.

Variable	Non-CR-GNB (57)	CR-GNB (49)	Total (106)	*p*-Value
*Therapeutic measures*				
CVC removal				
None (N (%))	17 (30%)	21 (43%)	38 (36%)	0.151
On suspicion (N (%))	22 (39%)	20 (41%)	42 (40%)	
On evidence (N (%))	18 (32%)	8 (16%)	26 (25%)	
ATB empiric Match (N (%))	27 (47%)	11 (23%)	38 (36%)	0.0080
ATB change	25 (44%)	22 (45%)	47 (44%)	0.9150
*Measures of outcome*				
Ventilator free days	19 (0–25)	0 (0–4)	4 (0–22)	0.0000
ICU LOS (days)	19 (8–37)	25 (13–44)	21 (11–40)	0.2631
In-hospital LOS (days)	42 (24–64)	47 (24–75)	44 (24–73)	0.5180
Days before ICU	3 (1–13)	3 (0–16)	3 (0–15)	0.9083
In-hospital death (N (%))	18 (32%)	24 (49%)	42 (40%)	0.0680
28-day death (N (%))	10 (18%)	17 (35%)	27 (26%)	0.0430
Death cause				
Infection (N (%))	15 (88%)	17 (74%)	32 (80%)	0.263
Other (N (%))	2 (12%)	6 (26%)	8 (20%)	

**Table 4 pathogens-11-00718-t004:** Uni- and Multi-variate logistic regressions of risk factors for CR-GNB aetiology for BSI. List of abbreviations: CR-GNB, carbapenem-resistant Gram-negative bacteria; CV, Cardiac and/or Vascular; ICU, Intensive Care Unit; VAP, Ventilator-Associated Pneumonia; VAT, Ventilator-Associated Tracheobronchitis; UT, Urinary Tract.

Dependent Variable	Independent Variables	Univariate					Multivariate				
		OR	SE	Z	CI 95%	*p*	OR	SE	Z	CI 95%	*p*
CR-GNB-**BSI+**	APACHE score	1.09	0.04	2.37	1.02–1.17	0.018					
	SAPS II score	1.04	0.02	2.50	1.01–1.08	0.012	1.04	0.02	2.06	1.00–1.09	0.039
	SOFA admission	1.14	0.07	2.14	1.01–1.29	0.033					
	Parenteral nutrition	2.50	1.05	2.18	1.10–5.71	0.029	1.14	0.60	0.24	0.40–3.22	0.808
	CV surgery before ICU admission	2.44	0.98	2.22	1.11–5.36	0.026	3.34	1.74	2.30	1.20–9.29	0.021
	Sepsis	4.84	2.49	3.08	1.78–13.24	0.002	1.97	1.29	1.03	0.54–7.13	0.302
	Septic shock	2.27	0.93	1.99	1.01–5.07	0.046					
	VAP	4.52	2.26	3.03	1.70–12.03	0.002					
	VAT	4.52	2.37	2.87	1.61–12.65	0.004					
	UT colonization	4.62	3.19	2.21	1.19–17.88	0.027					
	Gastro-intestinal colonization	33.88	22.52	5.30	9.21–124.69	0.000	10.58	5.90	4.23	3.59–31.55	0.000

## Data Availability

Data are available upon reasonable request to the corresponding author.

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
