# Peer review of "Bloodstream Infections Caused by Carbapenem-Resistant Pathogens in Intensive Care Units: Risk Factors Analysis and Proposal of a Prognostic Score"

_pathogens, 2022, doi:10.3390/pathogens11070718_

Round 1

Reviewer 1 Report

The authors have conducted interesting and informative research that can add useful information to the existing data on carbapenem-resistant pathogens in Intensive Care Units. The manuscript is well written. Data presentation and the methods used for the generation of the data are reasonable.

Major comments

Lies 163-165” please write the culture media that were used for microbiological culture and the method that was used for confirmation of carbapenem resistance.

Line 209 “Microbiological isolates (n= 182) are shown in Figure 1.”: please write the species of 182 isolates. Moreover, authors should provide a table presenting Gram negative ESBL-GNB: Extended Spectrum Beta-Lactamase Gram-Negative Bacteria  (n=10); GNB: non-multi drug resistant Gram-Negative Bacteria (n=29); MR-GP: Methicillin-Resistant Gram-Positive Bacteria (n=39); MS-GP: Methicillin-Sensitive Gram-Positive Bacteria (n=34); and Yeasts (n=34).

Minor comments

Line 18 “antibiotic-multi-resistant pathogens” should be: multidrug resistant pathogens

Line 40: Gram negative

Line 94 “ Scienza’ University Hospital in Turin (Italy)” should be: Scienza’ University Hospital in Turin, Italy.

Line 103”ECDC criteria” and L374 “MALDI-TOF “: please write in full at the first mention

Lines 105-106 and in the references section: bacterial species should be italized. Also, Line 183 “p” should be italic

Lines 173-174,  337: A. baumannii, K. pneumoniae, P. aeruginosa

Tables 1& 2: all abbreviations should be in the footnote of the table, not in the title

Author Response

Pathogens

Bloodstream infections caused by carbapenem resistant pathogens in Intensive Care Units: risk factors analysis and proposal of a prognostic score

Reviewer 1

The authors have conducted interesting and informative research that can add useful information to the existing data on carbapenem-resistant pathogens in Intensive Care Units. The manuscript is well written. Data presentation and the methods used for the generation of the data are reasonable.

We thank the Rev. 1 for his overall comment and for appreciating our work.

Major comments

Lies 163-165” please write the culture media that were used for microbiological culture and the method that was used for confirmation of carbapenem resistance.

Thank you for pointing out this aspect. we have introduced two more paragraphs in the methods section, explaining in detail what is required.

“Suspected GNB colonies grown on MacConkey II Agar (BD, USA) from tracheal and urinary cultures were identified and tested for CR using the semi-automated Microscan Walkaway 96 plus System (Beckman Coulter, USA) with Neg Combo 83 (Beckman Coulter) panels and a complete antimicrobial susceptibility test (AST) was provided. GNB isolated on selective solid media, Brilliance CRE AGAR (Oxoid, UK), were identified by MALDI-TOF VITEK MS (bioMérieux, France) and CR was confirmed by meropenem and imipenem E-test (bioMérieux).

BCs are performed in case of clinical suspicion of infection. BCs were incubated using the automated BC system BACT/ALERT 3D (bioMérieux). Positive BC were processed according to the laboratory routine workflow as previously described.  Briefly, BC were seeded on the appropriate solid medium after Gram staining results (Columbia Agar with 5% Sheep Blood, MacConkey II Agar, Columbia CNA Agar with 5% Sheep Blood, Chocolate Agar; Becton–Dickinson, USA) and isolated colonies after overnight incubation were tested for identification and AST both on Microscan Walkaway 96 plus System (Beckman Coulter). The minimum inhibitory concentrations results were interpreted according to the EUCAST clinical breakpoints for all tested antibiotics.”

Line 209 “Microbiological isolates (n= 182) are shown in Figure 1.”: please write the species of 182 isolates. Moreover, authors should provide a table presenting Gram negative ESBL-GNB: Extended Spectrum Beta-Lactamase Gram-Negative Bacteria  (n=10); GNB: non-multi drug resistant Gram-Negative Bacteria (n=29); MR-GP: Methicillin-Resistant Gram-Positive Bacteria (n=39); MS-GP: Methicillin-Sensitive Gram-Positive Bacteria (n=34); and Yeasts (n=34).

As requested by Rev.1, we have further detailed the microbiological isolates (n = 182) in a new table (number 1 in the order of appearance in the text).

Table 1: Microbiological isolates species in the overall population

Category

Species

N (overall %)

CR-GNB

K. pneumoniae

A. baumannii

P. aeruginosa

50 (27.5%)

8 (4.4%)

4 (2.2%)

ESBL-GNB

E. coli

K. pneumoniae

P. mirabilis

7 (3.9%)

1 (0.6%)

2 (1.1%)

GNB

E. coli

K. pneumoniae

K. oxytoca

E. aerogenes

S. marcescens

M. morganii

P. aeruginosa

S. maltophila

C. koseri

H. alvei

A. xylosoxidans

E. cloacae

A. baumannii

1 (0.6)%

1 (0.6)%

3 (1.7%)

1 (0.6)%

7 (3.9%)

3 (1.7%)

5 (2.8%)

1 (0.6)%

1 (0.6)%

1 (0.6)%

1 (0.6)%

3 (1.7%)

1 (0.6)%

MR-GP

S. epidermidis

S. aureus

S. haemolitycus

35 (19.2%)

2 (1.1%)

2 (1.1%)

MS-GP

E. faecalis

E. faecium

S. epidermidis

S. aureus

S. haemolitycus

S. capitis

S. simulans

G. sanguinis

S. cohnii

B. casei

6 (3.3%)

8 (4.4%)

9 (5.0%)

4 (2.2%)

1 (0.6%)

2 (1.1%)

1 (0.6%)

1 (0.6%)

1 (0.6%)

1 (0.6%)

Yeasts

C. albicans

C. glabrata

C. parapsilosis

3 (1.7%)

4 (2.2%)

1 (0.6%)

Total

182 (100%)

List of abbreviations: CR-GNB: Carbapenem Resistant Gram-Negative Bacteria; ESBL-GNB: Extended Spectrum Beta-Lactamase Gram-Negative Bacteria; GNB: non-multi drug resistant Gram-Negative Bacteria; MR-GP: Methicillin-Resistant Gram-Positive Bacteria; MS-GP: Methicillin-Sensitive Gram-Positive Bacteria; Yeasts.

Minor comments:

all minor comment have been addressed in the text.

Reviewer 2 Report

The manuscript includes an exhaustive analysis of the risk factors addressed in the study. Statistical methods and conclusions are consistent. The limitations of the research are appropriately marked.
It only has minimal spelling details to review. Those details are highlighted in yellow in the attached file.

Author Response

Pathogens

Bloodstream infections caused by carbapenem resistant pathogens in Intensive Care Units: risk factors analysis and proposal of a prognostic score

Reviewer 2

The manuscript includes an exhaustive analysis of the risk factors addressed in the study. Statistical methods and conclusions are consistent. The limitations of the research are appropriately marked.

We thank the Rev. 2 for his overall comment and for appreciating our work.

It only has minimal spelling details to review. Those details are highlighted in yellow in the attached file.

Thanks for the careful revision. All the points highlighted have been corrected in the text according to the indications.

Reviewer 3 Report

The manuscript is well-written and interesting, focused on a very important topic. Few minor corrections:

- All species names must be in italics

- Lines 114-115: Provide references for the severity scoring systems mentioned

- Lines 363-366: This section can be expanded a bit. What settings do you mean? What non-BSI cases do you mean?

- The manuscript is too focused on the Italian population and cardiovascular disease patients. Are there any studies from other countries and other patients? Is there agreement with your results?

Author Response

Pathogens

Bloodstream infections caused by carbapenem resistant pathogens in Intensive Care Units: risk factors analysis and proposal of a prognostic score

Reviewer 3

The manuscript is well-written and interesting, focused on a very important topic. Few minor corrections:

We thank the Rev. 3 for his overall comment and for appreciating our work.

- All species names must be in italics

Thanks for the careful revision. All the species names have been corrected in the text according to the indications.

- Lines 114-115: Provide references for the severity scoring systems mentioned

Thanks for this comment, we have added bibliographic references.

- Lines 363-366: This section can be expanded a bit. What settings do you mean? What non-BSI cases do you mean?

Thank you for raising this point. We have better specified what is requested and expressed the paragraph in a new form.

Several scores have been proposed in order to estimate the risk of developing infections by multi-resistant pathogens [18, 19, 22, 26, 27, 37]. However, these scores were focused either on different settings from ICU, or not only on BSI or on specific subgroups of patients.

For instance, Giannella et al (19-20) investigated the risk in liver transplant recipients colonized with carbapenem-resistant Enterobacteriaceae; Tumbarello et. al (22) identified risk factors specifically associated with K.pneumoniae KPC infection – regardless on the site- in all hospitalized patients, while Vadesuvan et al. (26) proposed a simple but effective score to predict nosocomial resistant Gram-Negative Bacilli infections among ICU patients, already validated in an external cohort (27).

- The manuscript is too focused on the Italian population and cardiovascular disease patients. Are there any studies from other countries and other patients? Is there agreement with your results?

Thanks for the comment. Certainly, our work refers to the Italian situation, as a comparison. On the other hand, the cardiovascular patients are described only referring to the results obtained on our cohort, as in the literature there are no great references in this sense. However, we underline that the references used as a comparison derive mainly from ECDC data, with European, and not only Italian, comparators. In general, on the whole, our data are in line with the literature, both regarding incidence, both with respect to the risk factors identified, with some peculiarities that have been discussed. Moreover, the bibliography - which has been extensively revised for past errors, of which we apologize - has been expanded in this sense. In general, as also requested by other reviewers, we have made various changes in the discussion, and we hope that the new version will overcome the highlighted limitations.

Reviewer 4 Report

Due to the growing prevalence of CR issue, the authors proposed a predictive prognostic model to help the clinicians identifying the patients with serve infections in ICU, giving the most appropriate empiric treatment, and ameliorating patient’s outcome. However, the authors presented the article in a careless way, with many format errors, redundant words, and inappropriate paragraphs. Although I provide some below (in minor comments), still please see carefully through the whole article.

Major comments

1.      Was any concurrent antibiotic therapy used for patients (colistin, aminoglycosides and so on)? It seems the patients were treated by carbapenems and/or quinolones. If yes, please describe or discuss; if not collected, please listed in limitation.

2.      Line 204-206. What percentage? There is no related data seeing in Table 1.

3.      Please transform Figure 1 to a table to provide detailed distribution of 182 isolates from 158 episodes in 106 patients.

4.      Result section. Please describe more to help understand data/model, especially Figures 2 and 3. It’s not easy to read in the current form. Also, in line 207-209, a total number should be described first, and then the distribution inside. (e.g. Microbiological isolates... shown in Figure 1. BSI were… … … by CR-P. aeruginosa (2; 4%))

5.      Please rephrase the WHOLE discussion section. It’s hard to read in the current type. A prospect a paragraph, including literature reviews and your finding, and please make them all in the same order (yours->reviews or reviews -> yours). DO NOT SEPARATE YOUR FINDING AND REVIEWS INTO TWO PARAGRAPHS!!

6.      Please make conclusion more concise (in 5-8 lines, it’s almost double now)

Minor comments

1.      E-mail on Affiliation was not annotated. Please find the writing rules in MDPI.

2.      Please make the main text align left and right.

3.      No need for subheadings (intro, obj, meth, result, conclusion) in abstract.

4.      Bacterial spp. name should be ITALIC! Please check ALL OVER the article including abstract.

5.      Keywords. Carbapenem antibiotics (please remove the word “antibiotics”); antimicrobial drug resistance (please remover the word “drug”).

6.      Introduction could be condensed into 3 paragraphs. Line 44-62: para1; Line 63-76: para 2; Line 77-89: para 3.

7.      Collected variables described in lines 109-138 could be list as a table to help reading.

8.      Tables 1 and 2. Shouldn’t the MDR be revise into CR?

9.      The decimals in tables were represented in different way (sometimes dot/point; sometimes comma) please make them uniformly in dot/point.

10.  There should be a space before every bracket (Such as table 1: carbapenem admission (N (%)) and others in tables 1 and 2, please check all of them).

11.  Line 263. Space after semicolon. (17; 74%)

12.  Table 3. Please check the format (row width) to make the values in the same column. (both p values; Z values in uni analysis)

13.  Figure 2. Poor resolution. Please provide a clearer version.

14.  Please recheck Reference formats (font, size, words should be italic, etc.)

Author Response

Pathogens

Bloodstream infections caused by carbapenem resistant pathogens in Intensive Care Units: risk factors analysis and proposal of a prognostic score

Reviewer 4

Due to the growing prevalence of CR issue, the authors proposed a predictive prognostic model to help the clinicians identifying the patients with serve infections in ICU, giving the most appropriate empiric treatment, and ameliorating patient’s outcome. However, the authors presented the article in a careless way, with many format errors, redundant words, and inappropriate paragraphs. Although I provide some below (in minor comments), still please see carefully through the whole article.

Thanks to the reviewer for his extended revision.

We are very sorry for this comment, we had absolutely no intention of presenting our work "in a careless way". However, we believe, thanks to the important work of the other reviewers 1, 2, 3 and their comments, as well as the ideas provided by the rev. 4, that we have improved the initial work. We confirm that the initial draft has been amply modified as a whole, and all the insights provided have been followed.

Below, the point responses to comments.

Major comments

  1. Was any concurrent antibiotic therapy used for patients (colistin, aminoglycosides and so on)? It seems the patients were treated by carbapenems and/or quinolones. If yes, please describe or discuss; if not collected, please listed in limitation.

Thank you for pointing out this aspect. In reality, the choice to deepen the previous antibiotic therapy was deliberately focused on carbapenems and fluoroquinolones, recognized as major selectors of pathogens difficult to treat according to the current literature. the data is not intended to describe prescriptive habits in itself (which would already be influenced by local ecology), but rather to define whether and to what extent the use of CP and FQ may represent a risk factor for the development of subsequent antibiotic resistance. We further clarified the point in the discussion.

With regard to previous antibiotic therapies, there was no correlation between the duration of antimicrobial therapy, nor between the administration of antibiotics known to select more resistances, such as carbapenems or fluoroquinolones, and the carbapenem-resistant BSI.

In consideration of the findings and of your revision, however, we noted in the limitations that the data relating to the previous antibiotic therapies of the patients studied were not available.

Regarding the impact of previous antimicrobial use on the possibility to develop CR-GNB strains, although neither carbapenems nor fluoroquinolones seem to play a causative role, it was not possible to fully collect data on other antibiotic therapies.

  1. Line 204-206. What percentage? There is no related data seeing in Table 1.

We apologize for this error, due to an incomplete revision of the table (in the first version, table 1; in the revised version, table 2). In fact, some lines were missing with the data that have been described and to which the reviewer refers. In the new version these lines have been reinstated and contain what is requested.

Variable

non-MDR CR-GNB (57)

MDR CR-GNB (49)

Total (106)

p-Value

Demographics

Age (years)

64 (55 – 73)

65 (53 – 73)

64 (54 – 73)

0.7998

Gender M (N(%)) (N (%))

38 (67%)

29 (59%)

67 (63%)

0.4260

ICU scoring systems

APACHE score

15 (12 – 20)

19 (15 – 23)

18 (14 – 21)

0.0121

SAPS II score

39 (28 – 49)

46 (38 – 54)

42 (33 – 52)

0.0101

SOFA admission

8 (5 – 9)

9 (7 – 11)

8 (5 – 10)

0.0305

CHARLSON index

4 (2 – 6)

4 (3 – 6)

4 (3 – 6)

0.4239

Comorbidities

Main diagnosis

Shock (N (%))

Cardiovascular (N (%))

Respiratory (N (%))

Abdominal (N (%))

Other (N (%))

19 (33%)

7 (12%)

18 (32%)

7 (12%)

6 (11%)

18 (37%)

8 (16%)

11 (23%)

7 (14%)

5 (10%)

37 (35%)

15 (14%)

29 (28%)

14 (13%)

11 (10%)

0.8650

Sepsis (N (%))

34 (60%)

43 (88%)

77 (73%)

0.0010

Septic shock (N (%))

16 (28%)

23 (47%)

39 (37%)

0.0450

Episodes of BSI

Single

46 (81%)

30 (61%)

76 (72%)

0.0260

Multiple

non-CR-GNB first

          CR-GNB first

11 (19%)

16 (33%)

3 (6%)

27 (26%)

3 (3%)

Potential source of BSI

VAT (N (%))

6 (11%)

17 (35%)

23 (22%)

0.0030

UT colonization (N (%))

3 (5%)

10 (20%)

13 (12%)

0.0340

Rectal colonization (N (%))

3 (5%)

32 (65%)

35 (33%)

0.0000

CLABSI (N (%))

46 (81%)

36 (74%)

82 (77%)

0.3570

Vital function support

MV (N (%))

46 (81%)

42 (86%)

88 (83%)

0.4930

Tracheostomy (N (%))

10 (18%)

20 (41%)

30 (30%)

0.0080

ECLS (N (%))

5 (9%)

10 (20%)

15 (14%)

0.1010

ECLS (days)

0 (0 – 0)

0 (0 – 0)

0 (0 – 0)

0.1471

RRT (N (%))

11 (19%)

14 (29%)

25 (24%)

0.2620

RRT (days)

0 (0 – 0)

0 (0 – 2)

0 (0 – 0)

0.2111

Parenteral nutrition (N (%))

14 (25%)

22 (45%)

36 (34%)

0.0280

LVAD (N (%))

5 (9%)

5 (10%)

10 (9%)

1.0000

PM/ICD (N (%))

9 (16%)

7 (14%)

16 (15%)

0.8290

Prior ATB therapy

Duration of ATB therapy

None

< 7 days

7-14 days

> 14 days

44 (77.19 77%)

6 (10.52 11%)

6 (10.52 11%)

1 (1.75 2%)

35 (71.42 71%)

7 (14.28 14%)

4 (8.16 8%)

3 (6.12 6%)

79 (74.52 75%)

13 (12.26 12%)

10 (9.43 9%)

4 (3.77 4%)

0.6020

Carbapenem before ICU admission (N (%))

6 (10.52 11%)

8 (16.32 16%)

14 (13.2 13%)

0.3790

Fluoroquinolone before ICU admission (N (%))

10 (17.54 18%)

9 (18.36 18%)

19 (17.9 18%)

0.9120

  1. Please transform Figure 1 to a table to provide detailed distribution of 182 isolates from 158 episodes in 106 patients.

As required by rev. 1, we have integrated the figure with the related table (see new Table 1 in the text). The data relating to the isolates are thus complete.

Table 1: Microbiological isolates species in the overall population

Category

Species

N (overall %)

CR-GNB

K. pneumoniae

A. baumannii

P. aeruginosa

50 (27.5%)

8 (4.4%)

4 (2.2%)

ESBL-GNB

E. coli

K. pneumoniae

P. mirabilis

7 (3.9%)

1 (0.6%)

2 (1.1%)

GNB

E. coli

K. pneumoniae

K. oxytoca

E. aerogenes

S. marcescens

M. morganii

P. aeruginosa

S. maltophila

C. koseri

H. alvei

A. xylosoxidans

E. cloacae

A. baumannii

1 (0.6)%

1 (0.6)%

3 (1.7%)

1 (0.6)%

7 (3.9%)

3 (1.7%)

5 (2.8%)

1 (0.6)%

1 (0.6)%

1 (0.6)%

1 (0.6)%

3 (1.7%)

1 (0.6)%

MR-GP

S. epidermidis

S. aureus

S. haemolitycus

35 (19.2%)

2 (1.1%)

2 (1.1%)

MS-GP

E. faecalis

E. faecium

S. epidermidis

S. aureus

S. haemolitycus

S. capitis

S. simulans

G. sanguinis

S. cohnii

B. casei

6 (3.3%)

8 (4.4%)

9 (5.0%)

4 (2.2%)

1 (0.6%)

2 (1.1%)

1 (0.6%)

1 (0.6%)

1 (0.6%)

1 (0.6%)

Yeasts

C. albicans

C. glabrata

C. parapsilosis

3 (1.7%)

4 (2.2%)

1 (0.6%)

Total

182 (100%)

List of abbreviations: CR-GNB: Carbapenem Resistant Gram-Negative Bacteria; ESBL-GNB: Extended Spectrum Beta-Lactamase Gram-Negative Bacteria; GNB: non-multi drug resistant Gram-Negative Bacteria; MR-GP: Methicillin-Resistant Gram-Positive Bacteria; MS-GP: Methicillin-Sensitive Gram-Positive Bacteria; Yeasts.

  1. Result section. Please describe more to help understand data/model, especially Figures 2 and 3. It’s not easy to read in the current form. Also, in line 207-209, a total number should be described first, and then the distribution inside. (e.g. Microbiological isolates... shown in Figure 1. BSI were… … … by CR-P. aeruginosa (2; 4%))

Thank you for raising this point. We have inserted a further paragraph to explain, as required, the model and its use. Moreover, we corrected the total number/distribution as suggested.

“Figure 2 is the nomogram built using the variables included in the model. Their presence or absence allow to obtain an overall probability of developing CR-GNB BSI.

Such a model has been calibrated based on a 1000 repetitions bootstrap process (see Methods), and Figure 3 shows results of it. The prediction model has some divergences from the ideal prediction (i.e., the dotted line) but their entity is negligible at about 40% and 90% probability of CR-GNB BSI.”

  1. Please rephrase the WHOLE discussion section. It’s hard to read in the current type. A prospect a paragraph, including literature reviews and your finding, and please make them all in the same order (yours->reviews or reviews -> yours). DO NOT SEPARATE YOUR FINDING AND REVIEWS INTO TWO PARAGRAPHS!!

Thank you for your suggestion. The whole discussion has been extensively revised, both in terms of content and in the order of the paragraphs. As suggested, we presented firstly our data, followed by the literature comment. We hope that this new version will be clearer and respond to the reviewer's requests.

  1. Please make conclusion more concise (in 5-8 lines, it’s almost double now)

Thanks for this suggestion. We have re-formulated the conclusions, which are now much more concise.

About a half of patients enrolled in our study developed CR-GNB BSI, in clear superiority if compared to the literature and with a strong impact on patient survival, as these pathogens were associated with higher mortality and lower efficacy of initial empirical therapy. Specific risk factors - clinical patient severity, presence of sepsis, concomitance of respiratory infections and gastrointestinal colonization- are in line with the literature. Our clinical easy, fast-performing prediction tool could enable a better targeting of the early antimicrobial treatment for BSI, maintaining an antimicrobial stewardship perspective also in the ICU context with its elevated incidence of CR-resistance. Further multicentric and prospective studies are needed to confirm our data and to validate the model and better define the impact of each risk factor.

Minor comments

  1. E-mail on Affiliation was not annotated. Please find the writing rules in MDPI. Thanks for the comment. We corrected accordingly.
  2. Please make the main text align left and right. Thanks for the comment. We corrected accordingly.
  3. No need for subheadings (intro, obj, meth, result, conclusion) in abstract. Thanks for the comment. We corrected accordingly.
  4. Bacterial spp. name should be ITALIC! Please check ALL OVER the article including abstract. Thanks for the comment. We corrected accordingly.
  5. Keywords. Carbapenem antibiotics (please remove the word “antibiotics”); antimicrobial drug resistance (please remover the word “drug”). Thanks for the comment. We corrected accordingly.
  6. Introduction could be condensed into 3 paragraphs. Line 44-62: para1; Line 63-76: para 2; Line 77-89: para 3. Thanks for the comment. We corrected accordingly.
  7. Collected variables described in lines 109-138 could be list as a table to help reading.

Thank you for your suggestion. We have revised the paragraph, limiting the number of variables listed, in order to increase the ease of reading, without having to add an additional table.

  1. Tables 1 and 2. Shouldn’t the MDR be revise into CR? Thanks for the comment, it was a mistake. We corrected accordingly in each table.
  2. The decimals in tables were represented in different way (sometimes dot/point; sometimes comma) please make them uniformly in dot/point. Thanks for the comment. We corrected accordingly.
  3. There should be a space before every bracket (Such as table 1: carbapenem admission (N (%)) and others in tables 1 and 2, please check all of them). Thanks for the comment. We corrected accordingly.
  4. Line 263. Space after semicolon. (17; 74%) Thanks for the comment. We corrected accordingly.
  5. Table 3. Please check the format (row width) to make the values in the same column. (both p values; Z values in uni analysis).

Thanks for this point comment. We revised the table accordingly.

  1. Figure 2. Poor resolution. Please provide a clearer version.

We apologize for the inconvenience. We are sending new files separately so that they are of better quality.

  1. Please recheck Reference formats (font, size, words should be italic, etc. Thanks for the comment. We corrected accordingly.

Round 2

Reviewer 1 Report

I really appreciate the modification made by the authors. I have minor further comments as follows:

1- lines 167-168 “The minimum inhibitory concentrations results were interpreted according to the EUCAST clinical breakpoints for all tested antibiotics”: please write the reference of EUCAST.

2-Please write the MIC values of the tested antibiotics in the results section.

3-line 162 “as previously described [35], Briefly, BC…….” should be “as previously described [35]. Briefly, BC…….”

4-In table 1: please write “Number (overall %)” instead of “N (overall %)” and delete (%) from the column to avoid repetition as it is written in the heading of the table.

5-Lines 309-310: Gram-negative

6-References: all species should be italicized

Author Response

I really appreciate the modification made by the authors.

Thank you for your suggestions and for this comment.

I have minor further comments as follows:

1- lines 167-168 “The minimum inhibitory concentrations results were interpreted according to the EUCAST clinical breakpoints for all tested antibiotics”: please write the reference of EUCAST.

Thank you for this point, we added the reference in the bibliography.

2-Please write the MIC values of the tested antibiotics in the results section.

As requested, we added this sentence in the results section:

"In CR-GNB group, registered MIC were always >32 μg/mL when confirmed by E-test (bioMérieux) methods."

In addition, regarding MIC, we add also this sentence in the method section:

"The tested carbapenems minimum inhibitory concentrations (MIC) were 0.5-1 μg/mL for ertapenem, 1-8 μg/mL for imipenem, 0.12, 2, 8 μg/mL for meropenem with Neg Combo 83 (Beckman Coulter) panels and 0.002-32 μg/mL for both imipenem and meropenem with E-test (bioMérieux). For all tested antibiotics MIC values were interpreted according to the EUCAST clinical breakpoints. "

3-line 162 “as previously described [35], Briefly, BC…….” should be “as previously described [35]. Briefly, BC…….”

Thanks, we changed accordingly.

4-In table 1: please write “Number (overall %)” instead of “N (overall %)” and delete (%) from the column to avoid repetition as it is written in the heading of the table.

Thanks, we changed accordingly.

5-Lines 309-310: Gram-negative

Thanks, we changed accordingly.

6-References: all species should be italicized

Thanks, we changed accordingly.

Reviewer 4 Report

It's okay for pub

Author Response

Thank you very much for your help!

Kind regards,

Giorgia Montrucchio and co-authors